# Trends in the Frequency of Water and Heat Stress in Mid-Latitude North America since 1980

Arik Tashie 

Department of Civil, Construction, and Environmental Engineering, University of Alabama, Tuscaloosa, AL 35487, USA; tashi002@ua.edu

**Abstract:** The water-energy balance of many mid-latitude watersheds has been changing in recent decades due to global warming. These changes manifest themselves over both long timescales (e.g., hydrologic drought) and short timescales (e.g., agricultural drought) and may be ameliorated or exacerbated by vegetative response. We apply a Budyko framework to assess short-term response to long-term trends in water and heat stress (HS) across mid-latitude North America. Using high-resolution meteorological data and streamflow records, we calculate the frequency of HS every year since 1980 for every gaged watershed with adequate data ($n = 1528$). We find that HS has become more frequent in most watersheds in the western US, New England, and southeastern Canada. However, we find that HS has become less frequent in the Midwest and the relatively humid eastern US. By assessing the relationship between trends in HS frequency and proximate forcing variables (annual PPT, annual streamflow, minimum and maximum daily temperatures, actual evapotranspiration, and potential evapotranspiration), we find that these trends in HS frequency are primarily driven by meteorological forcings rather than vegetative response. Finally, we contextualize our findings within the Budyko framework, which assumes a landscape in equilibrium with its climate, with the implication that these trends in HS are only likely to be realized after local vegetation has adapted to new meteorological norms.

**Keywords:** climate change; water stress; Budyko



## 1. Background

Episodic drought and heat stress place major limitations on plant productivity worldwide [1]. Flash droughts and extreme episodic heat stress have been linked to major tree die offs around the world [2]. Similarly, acute water or heat stress in the early development of cereal crops impinges floral development and seed production, affecting the long-term productivity of many crops [3]. Identifying trends in the periodicity of these extreme events is essential for accounting for and potentially mitigating their effects on natural systems and agricultural productivity. However, while trends in long-term temperature averages and annual precipitation (PPT) are well documented [4,5], changes in the frequency of acute heat stress are less well understood. This analysis represents the first continental-scale trends analysis of acute heat stress that relies entirely on empirical data.

## 2. Introduction

Water and heat stress are typically quantified according to a specific temporal scale that defines the specific hydrologic effects of interest as well as the typical timescale for recovery [6]. For instance, meteorological drought describes a short-term deficit in rainfall in relation to long-term averages, and agricultural drought is defined by short-term, localized deficiencies in root zone soil moisture relative to vegetative demands [7]. Conversely, hydrologic drought occurs over longer timeframes and at a coarser spatial resolution. Since meteorological and agricultural drought depend on local, short timeframe meteorological variables, analysis of long-term trends in their frequency requires data that

are both highly temporally and spatially resolved and for which long-term records are available. Conversely, since the effects of hydrological drought are primarily realized on the scale of months, years, or decades, the data required for detection of trends in hydrological drought may be more coarsely resolved [8].

Common methods of calculating indices for each class of drought require varying degrees of data and computational complexity, creating a tradeoff between precision versus uncertainty and resolution [9]. Because in situ data are sparse in many regions, large-scale analysis must rely on remotely sensed climate data or climate reanalysis databases to some degree with the inherent tradeoff that these data sources are prone to greater uncertainty than are many in situ data products [9].

While most in situ data relevant to drought integrates at highly local scales (meters or tens of meters for soil moisture [10] or groundwater levels [11]), streamflow data effectively integrate all stores and fluxes at all points upstream of a gage [12]. Therefore, integrating high-temporal resolution, low uncertainty streamflow measurements with regional-scale climate data helps to minimize the biases and uncertainties inherent in climate databases while allowing the broad, regional-scale analyses that make their application so effective [13].

The Budyko curve has long been applied to describe annual average water-energy balance, the partitioning of evapotranspiration and streamflow, and transitions in overall climatic conditions [14,15]. In the Budyko framework, the fraction of annual precipitation (PPT) that is actually consumed by evapotranspiration (AET) is described in relation to total annual potential evapotranspiration (PET) and that fraction that is partitioned to streamflow (Q). Broadly, if the annual energy for PET in a stable landscape is greater than annual PPT, the annual AET approaches annual PPT; if the annual energy for PET is not sufficient, then annual AET approaches annual PPT. For most landscapes in a state of equilibrium, this balance between energy- and water-limits is generally predictable according to a simple equation (see Section 3: Methods). While many formulations of the Budyko curve exist, they generally follow the same pattern [16] (see Figure 1). An additional benefit of this framework is that it allows the approximation of AET (which is difficult to measure) using only PPT and PET (which are comparably easy to measure).

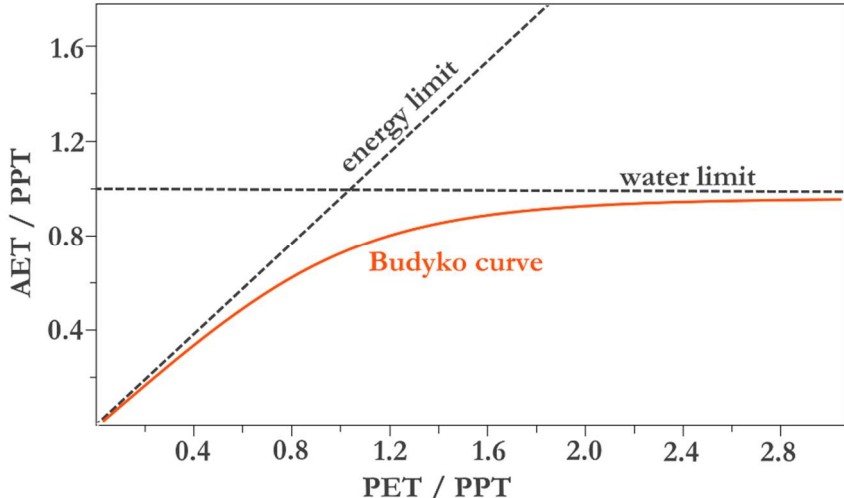

**Figure 1.** Schematic of a Budyko curve, illustrating the transition from an energy-limited to a water-limited system.

Application of the Budyko framework in small-scale analyses has confirmed that for landscapes in equilibrium, vegetation develops to maximize the use of PET (in energy-limited systems) and PPT (in water-limited systems) [17]. Thus, the placement of a watershed along a Budyko curve allows the estimation of the long-term average capacity of that watershed to evapotranspire in relation to average PPT. Therefore, sharp increases in PET

that substantially exceed average AET capacity represent episodes of water or heat stress (HS) that incorporate both meteorological forcings and vegetative feedbacks.

In this study, we calculate trends in the frequency of HS across mid-latitude North America over the past four decades. Specifically, we address the following questions:

1. Are episodes of acute HS becoming more or less frequent?
2. Are different regions experiencing similar or divergent trends in HS?
3. What are the proximate drivers of these trends in HS?

## 3. Methods

Calculation of the Budyko curve requires data for three key variables: (1) PET; (2) PPT; and (3) AET. PET and PPT can be estimated with purely meteorological data. However, estimates of annual average AET may be solved according to a basic mass balance equation that requires watershed streamflow (Q) data as well: AET = PPT − Q [18]. Long-term, high-quality Q data are available in the United States (US) from the US Geological Survey (USGS) and in Canada from the Water Data Archive.

We accessed Q data from all US and Canadian gages in the R programming language using the dataRetrieval package [19] and the tidyhat package [20]. We removed from all analysis all watersheds that were identified as having been significantly impacted by impoundments or other human activity (i.e., "nonreference" watersheds in the USGS data [21] and "regulated" watersheds in the Water Survey of Canada data [22]. We further removed all high latitude watersheds (defined as watersheds whose centroid was greater than 49° North) to focus on temperate watersheds that are more likely to experience high frequency HS than high latitude, energy-limited watersheds. We derived watershed boundaries from the HydroSHEDS data set at basin level 12 [23], with watersheds being defined as all level 12 basins upstream of the basin that intersects with the stream gage coordinates.

We assessed the quality of Q data according to the following criteria. First, we removed all records that were flagged by the USGS or Water Data Archive as having potential errors. To focus analysis on long-term climate trends and minimize the potential effects of multidecadal oscillations, we included for further analysis only gages with greater than 20 full years of record between the years 1980 and 2022, where a "full year" is defined as having non-zero streamflow (Q > 0) with gaps in the record for each month of the year. There remained 1528 watersheds with sufficiently high-quality data for analysis.

We derived climate data from the Daymet V3 data catalog [24], which provides the highest resolution (1 km$^2$) climatological data available for all of North America. Daymet provides daily estimates of PPT, minimum temperature (minT), maximum temperature (maxT), and day length (among several other climate variables not used in this analysis) beginning in 1980 and ending in 2022. For all relevant variables, we calculated a daily average watershed value by sampling each point within each watershed, then taking the mean.

Then, we calculated PET according to the Priestly-Taylor method [25] with albedo, terrestrial emissivity, aspect, slope, and the Priestly-Taylor constant set to constants of 0.18, 0.97, 0, 0, and 1.26, respectively. These calculations were carried out using the EcoHydRology package in R [26]. We calculated the aridity index (AI) as annual PET divided by annual PPT and AET as annual PPT minus annual Q. For each watershed for each year of record, we calculated a Budyko curve according to:

$$\frac{\text{AET}}{\text{PPT}} = \left[ \text{AI} \tanh \frac{1}{\text{AI}} (1 - \exp(-\text{AI})) \right]^{1/2} \tag{1}$$

where AI is the aridity index (PET/PPT) [14].

It should be noted that there are several functional forms of the Budyko equation as well as a variety of methods for calculating PET and AET. The choice to rely on Equation (1) (the Budyko equation in its original formulation) rather than one of the several other formulations was twofold: this formulation is among the most commonly applied in the

literature and the curve derived from it tends to plot between (i.e., an average of) the curves derived from other formulations [27]. However, the utility of alternative formulations lies in the fact that different watersheds with varying physioclimatic attributes may be better represented by alternative formulations. Therefore, the need to apply a consistent approach (i.e., Equation (1)) at the continental scale has the unfortunate result that the following analysis may be biased (or less well representative of) some watersheds.

For each watershed, we then calculated the episodes of HS, where an episode of HS is defined as each day when daily PET exceeds annual average AET by a factor of 2. Then, we summed the number of episodes of HS that occurred during each water year (October 1st–September 30th).

Because meteorological response to changes in climate tends to be highly nonlinear, susceptible to threshold response, and likely to include a number of statistical outliers, we chose to assess trends using the Theil–Sen slope [28,29] and Spearman's rank correlation test [30], which are more sensitive to detecting climate trends under these conditions [31,32]. For strength and directionality of trend, we relied on the Theil–Sen slope and on Spearman's rho, while for significance we relied on Spearman's $p$ value ($p < 0.1$).

To identify potential proximate drivers of trends in frequency of HS, we repeated this trends analysis on six relevant variables: annual total PPT, daily average minimum temperatures (minT), daily average maximum temperatures (maxT), annual total stream-flow (Q), the aridity index (AI = PET/PPT), and annual total observed evapotranspiration (AET = PPT − Q). For each of these variables, we again calculated watershed annual values from daily data. Then, we performed trends analysis using the Theil–Sen slope and Spearman's rank correlation test as described above.

## 4. Results

HS has become less frequent in a majority of mid-latitude North American gaged watersheds, with a general decrease in HS frequency in 59% of watersheds ($n = 909$) and general increase in HS frequency in 41% of watersheds ($n = 619$) (see Table 1). However, trends were significant ($p < 0.1$) in only 25% of watersheds ($n = 382$). When only significant trends are considered, then significant decreasing trends ($n = 277$) are more than twice as likely as significant increasing trends ($n = 105$).

**Table 1.** Strength of trends in HS frequency. All values are in units of days of HS per year per decade. Values were derived using Theil–Sen slope.

|  | 1st Quartile | Median | Mean | 3rd Quartile |
|---|---|---|---|---|
| Decreasing HS | −1.1 | −2.9 | −3.6 | −5.0 |
| Significantly Decreasing HS | −3.8 | −5.0 | −6.2 | −6.7 |
| Increasing HS | 0.7 | 1.6 | 2.6 | 3.1 |
| Significantly Increasing HS | 2.1 | 2.9 | 4.2 | 4.6 |

For watersheds with a significant negative trend in HS frequency, HS has been decreasing by about 5 days of HS per year every decade (1st Quartile = −3.8 days per year per decade; 3rd Quartile = −6.7 days per year per decade; Mean = −6.2 days per year per decade) (see Table 1). For watersheds with a significant positive trend in HS frequency, HS has been increasing by about 2.9 days of HS per year per decade (1st Quartile = 2.1 days per year per decade; 3rd Quartile = 4.6 days per year per decade; Mean = 4.2 days per year per decade).

The distribution of trends in HS exhibited extremely consistent patterns across mid-latitude North America (Figure 2). The vast majority of the western US, including the entirety of the Rocky Mountains and westward to the Pacific Ocean, exhibited a generally positive trend in the frequency of HS. These trends were especially consistent and more likely to be significant in the arid Southwest and in the humid Pacific Northwest. Most of mid-latitude Canada (i.e., southern Ontario, southern Quebec, New Brunswick, New-

foundland, and Nova Scotia) has also seen a general increase in HS frequency, although this pattern is somewhat less consistent than that seen in the western US. With some exceptions in New England, that vast majority of the rest of the US has seen a general decrease in HS frequency.

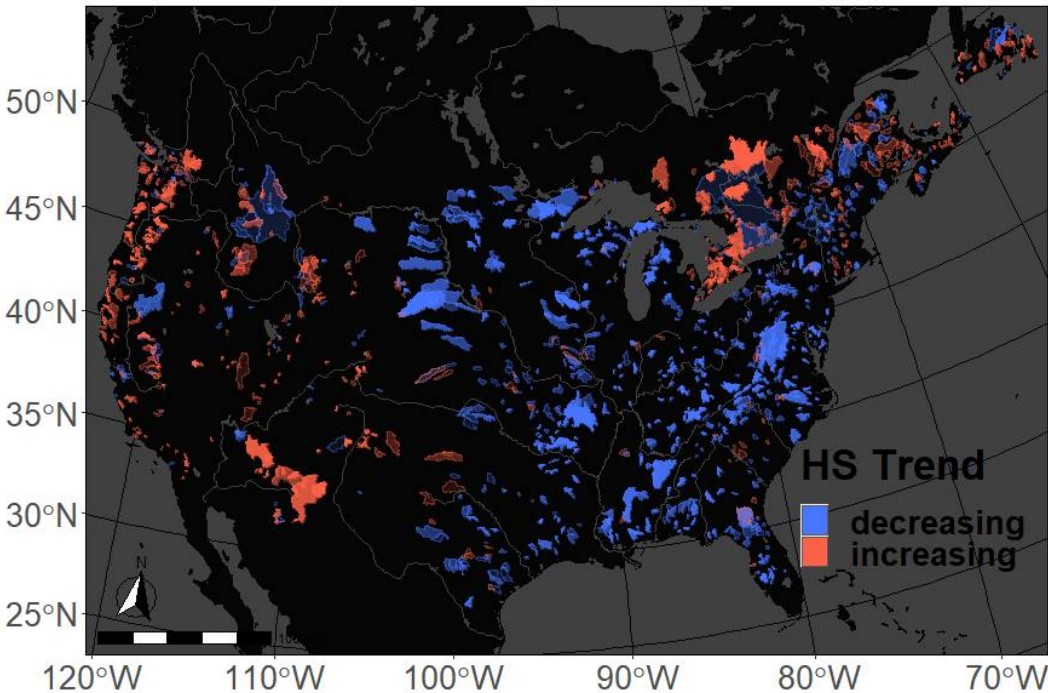

**Figure 2.** Trends in annual frequency of daily episodes of water and heat stress (HS). Watersheds exhibiting a significant trend ($p > 0.1$) are opaque while trends that are not significant are illustrated as semi-transparent. Increasing trends are in red and decreasing trends are in blue.

These regional patterns are also strongly reflected in the average trends across lines of longitude and latitude (Figure 3). At 24° N, HS trends are nearly universally negative. However, the likelihood of positive HS trends increases as one traces farther north, such that by around 45° N a majority of HS trends are negative. Patterns traced across lines of longitude, however, show a somewhat more complicated pattern. Along both coasts, HS trends tend to be positive. However, the interior of mid-latitude North America sees a reverse trend, with HS having become generally less frequent over the previous four decades. This reversal begins and ends at around 50° W and 105° W and is especially consistent from around 80° W to 100° W.

The six potential proximate drivers of trends in HS frequency we selected for this analysis show similarly strong regional patterns (Figure 4 and Table 2). PPT has increased in the vast majority of watersheds from 70° W to 100° W and has been more likely to decrease farther west and in southeastern Canada. MinT and MaxT have been generally increasing across most of mid-latitude North America, with a major exception in the Midwest which has seen a consistently declining MaxT over the previous 40 years. Despite these broad increases in temperature, the aridity index (AI) has actually been decreasing across most of the US east of 100° W, likely as a function of increased PPT in these regions. Both streamflow (Q) and AET have seen fewer significant trends, and both Q and AET have been more likely to decline farther to the south and west and more likely to increase farther north and east.

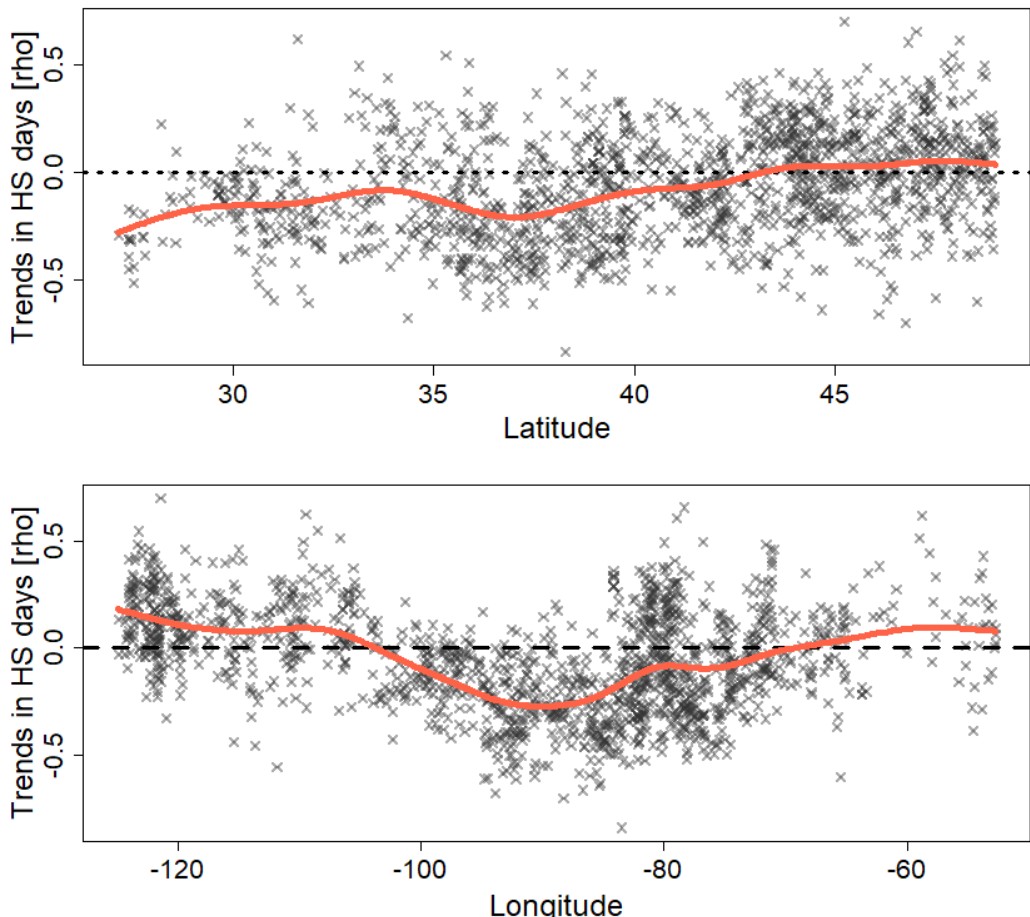

**Figure 3.** Relative trends in the annual frequency of daily episodes of water and heat stress (HS) by latitude (**top**) and longitude (**bottom**). The red line is a fitted smoothing spline with 10 degrees of freedom.

**Table 2.** Summary of the seven variables analyzed in this study.

|  | Min | 1st Quartile | Median | Mean | 3rd Quartile | Max |
|---|---|---|---|---|---|---|
| Heat Stress (HS) slope | −3.63 | −0.34 | −0.07 | −0.11 | 0.11 | 12.00 |
| Heat Stress (HS) rho | −0.84 | −0.24 | −0.06 | −0.06 | 0.12 | 0.70 |
| Heat Stress (HS) pval | 0.00 | 0.10 | 0.32 | 0.38 | 0.63 | 1.00 |
| PPT slope | −44.26 | −0.25 | 2.37 | 2.82 | 5.66 | 54.80 |
| PPT rho | −0.75 | −0.01 | 0.13 | 0.13 | 0.29 | 0.81 |
| PPT pval | 0.00 | 0.08 | 0.33 | 0.39 | 0.66 | 1.00 |
| Min Temp slope | −0.12 | 0.02 | 0.04 | 0.04 | 0.05 | 0.26 |
| Min Temp rho | −0.69 | 0.27 | 0.45 | 0.43 | 0.62 | 0.94 |
| Min Temp pval | 0.00 | 0.00 | 0.01 | 0.14 | 0.14 | 1.00 |
| Max Temp slope | −0.18 | 0.00 | 0.02 | 0.02 | 0.03 | 0.33 |
| Max Temp rho | −0.72 | 0.04 | 0.19 | 0.19 | 0.34 | 0.78 |
| Max Temp pval | 0.00 | 0.05 | 0.25 | 0.33 | 0.58 | 1.00 |
| Streamflow (Q) slope | −11.37 | −0.69 | 0.72 | 2.42 | 3.02 | 48.17 |
| Streamflow (Q) rho | −0.96 | −0.07 | 0.07 | 0.07 | 0.21 | 0.79 |
| Streamflow (Q) pval | 0.00 | 0.16 | 0.42 | 0.44 | 0.70 | 1.00 |

**Table 2.** *Cont.*

|  | Min | 1st Quartile | Median | Mean | 3rd Quartile | Max |
|---|---|---|---|---|---|---|
| Aridity Index slope | −0.20 | 0.00 | 0.00 | 0.00 | 0.00 | 0.30 |
| Aridity Index rho | −0.81 | −0.25 | −0.09 | −0.09 | 0.07 | 0.83 |
| Aridity Index pval | 0.00 | 0.10 | 0.35 | 0.40 | 0.66 | 1.00 |
| AET slope | −13.17 | −0.89 | −0.13 | 0.11 | 1.12 | 27.11 |
| AET rho | −0.75 | −0.18 | 0.00 | −0.01 | 0.16 | 0.91 |
| AET pval | 0.00 | 0.10 | 0.34 | 0.40 | 0.68 | 1.00 |

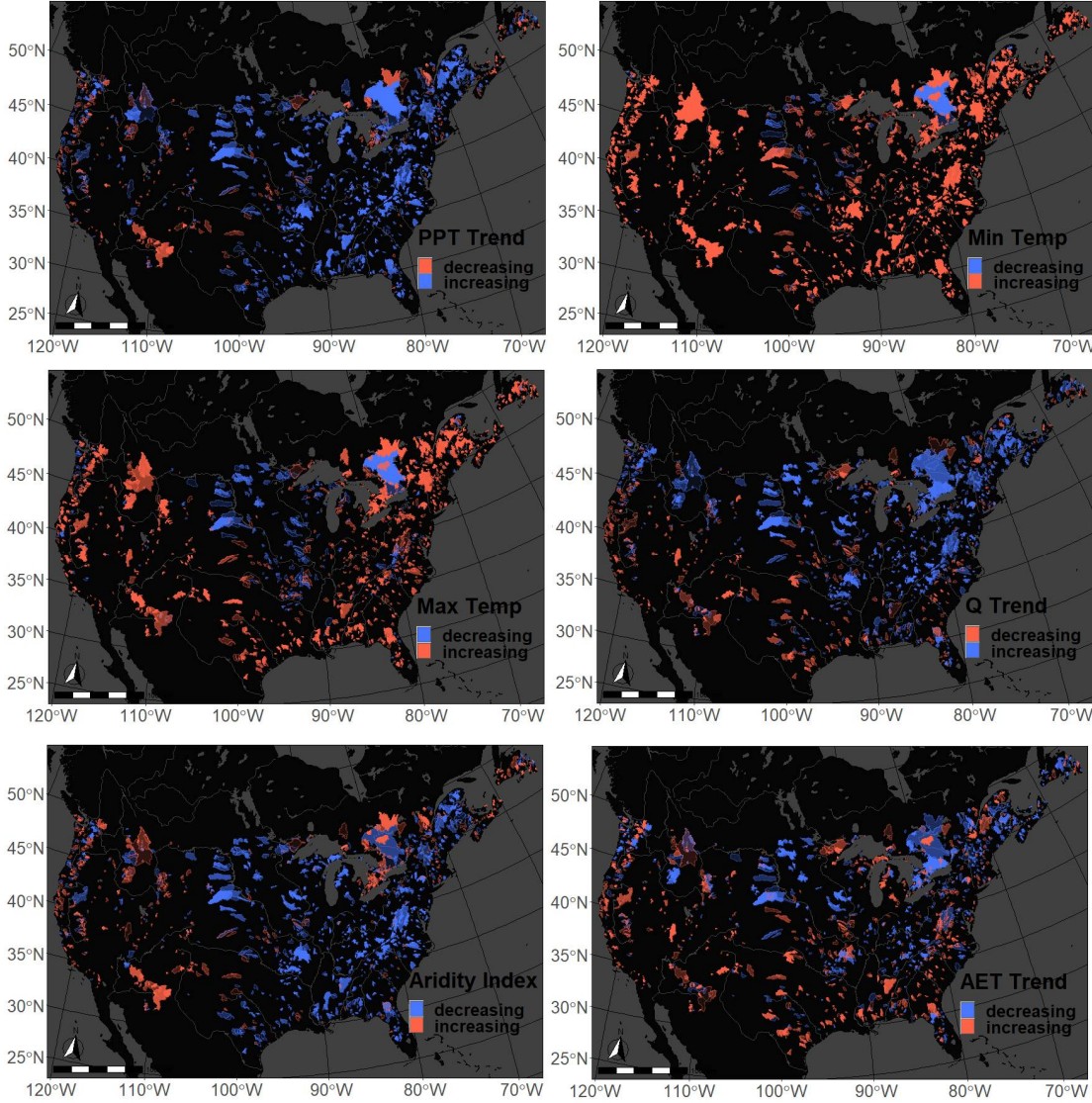

**Figure 4.** Trends in annual total precipitation (PPT—**top left**), average daily minimum temperature (Min Temp—**top right**), average daily maximum temperature (Max Temp—**middle left**), annual total streamflow (Q—**middle right**), annual aridity index (**bottom left**), and annual total evapotranspiration (AET—**bottom right**). Watersheds exhibiting a significant trend (*p* > 0.1) are opaque while trends that are not significant are illustrated as semi-transparent. Blue indicates increasing PPT and Q and decreasing Min Temp, Max Temp, Aridity Index, and AET. Red indicates decreasing PPT and Q and increasing Min Temp, Max Temp, Aridity Index, and AET.

## 5. Discussion

In the contiguous US, HS frequency has been increasing in the arid West and decreasing in the relatively humid eastern US. Indeed, the 100th Meridian that has traditionally been used to demarcate the water-limited West from the humid eastern US [33] also clearly demarcates the transition from watersheds with a high likelihood of increasing HS frequency and a high likelihood of decreasing HS frequency (Figure 2). These trends in exacerbating historical patterns (i.e., humid regions getting wetter and arid regions getting drier) aligns well with other studies illustrating the same trends in regard to soil moisture and aridity index over the previous decades [34].

However, these patterns do not persist in New England or across southeastern Canada, where HS frequency has been much more likely to increase than decrease since 1980 despite the aridity index (PET/PPT) in these regions being among the lowest in North America [35]. Similarly, in the extremely humid Coast Range and Cascade Range of the Pacific Northwest, HS frequency has been almost universally increasing.

Nonetheless, HS frequency strongly covaries with trends in total annual precipitation (PPT) (Figure 5). Regions that have seen increased PPT have seen a commiserate decline in the frequency of HS (r-squared of a linear model = 0.66). This relationship with increased water supply is much stronger than the weaker (but still significant) relationship with increased PET demands as indicated by trends in daily average minimum and maximum temperatures (MinT's r-squared of a linear model = 0.04; MaxT's r-squared of a linear model = 0.13). The greater control of MaxT relative to MinT is likely due both to its greater impact on PET and the fact that HS more strongly depends on changes in MaxT extremes.

HS frequency was also significantly correlated with trends in annual streamflow (Q), with increased Q correlating with decreased HS frequency (r-squared of a linear model = 0.19). However, trends in annual total AET were uncorrelated with HS frequency (r-squared of a linear model = 0.00). Given that the calculation of AET in the Budyko framework is the difference of annual total PPT and annual total Q, this strong correlation between HS frequency and PPT but weaker correlation with Q and lack of correlation with AET is somewhat surprising prima facie. That is, changes in the total water supply in a watershed (i.e., PPT) have been strongly correlated with HS trends, but the partitioning of that water (i.e., Q or AET) has been weakly correlated with HS trends. Therefore, HS trends have not been driven by changes in landcover or the adaptation of vegetation to new climate norms but have instead been primarily driven by changes in purely meteorological forcings (i.e., PPT and MaxT). This pattern is further borne out by the extremely strong relationship between HS trends and the aridity index (AI = PPT/PET; r-squared of a linear model = 0.76).

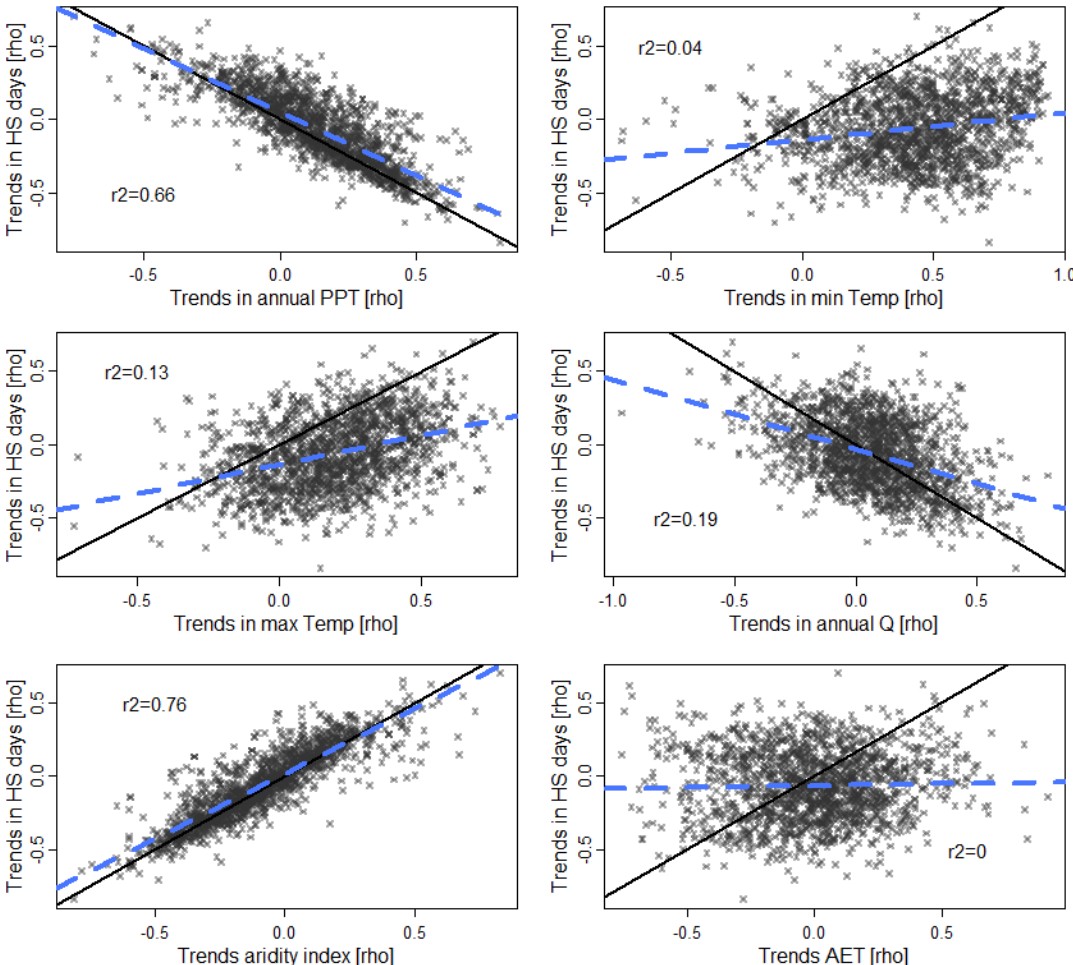

**Figure 5.** Relationship between frequency of water and heat stress (HS, on *y*-axes) and trends in PPT (**top left**), minimum temperature (**top right**), maximum temperature (**middle left**), annual streamflow (Q) (**middle right**), the aridity index (**bottom left**), and AET (**bottom right**). Solid black lines indicate a one-to-one line, while dashed blue lines represent a linear regression. R-squared values of that linear regression are in the corner of each plot.

## 6. Contextualization

That HS frequency has declined across half of mid-latitude North America since the 1980s despite the general increase in temperature over the same period is a somewhat surprising result. However, it is important to recall that the Budyko equation relates the balance between water and energy inputs over long timescales in landscapes where vegetation is in a state of equilibrium [36]. The underlying assumption is that vegetation competes to maximize the utilization of limiting physical inputs, i.e., water and energy, and that over time the species that are best able to maximize the use of these limiting resources will dominate less efficient utilizers of these resources [37].

Indeed, deviations from the Budyko are commonly used to investigate potential changes in land use and landcover [38]. Similarly, long-term trends in climate that occur at rates faster than local vegetation is capable of acclimating can also shift actual water-energy partitioning off the conceptual Budyko curve. The degree to which human-induced changes in landcover and climate affect this partitioning, and the degree to which these shifts can be quantified, is an important application of Budyko analysis in many contexts but is beyond the scope of this study. Therefore, increased (or decreased) HS frequency in our analysis should be contextualized as the potential response across a landscape after local vegetation has become fully acclimatized to the new climatic regime. However, any transition away from current (or past) conditions to a new climatic regime will be inherently

disruptive to the local vegetation over the intervening timeframe. Furthermore, agricultural production that does not shift to meet the new climatic regime is likely to become increasingly inefficient as productivity becomes increasingly incapable of maximizing the use of water and energy inputs. Similarly, it is important to note that the threshold above which plants experience acute heat stress varies among individual plant species. While the threshold of PET > 2 × AET applied in this analysis may be representative of typical vegetative response, the actual value is somewhat arbitrary and may not be representative of the response of individual plant species.

Combined with the result that in our analysis we found no correlation between trends in AET and HS frequency, but a strong correlation between HS frequency and both PPT and AI, it is likely the changes in water-energy budgets in mid-latitude North America since 1980 have been primarily meteorologically driven and vegetation driven. In watersheds undergoing this shift in water-energy balance, vegetation has likely not yet adapted to maximize the utilization of these limiting inputs, as evidenced by the greater general increase in Q versus increases in AET since 1980 (see Table 2). These trends in HS frequency, in part, represent a shift of the conceptual Budyko curve under highly dynamic climatic conditions. Therefore, even in watersheds that have seen a decrease in HS frequency according to the Budyko framework, vegetation is likely to have experienced an increase in meteorological and agricultural drought, as documented in previous studies [38].

It is also important to note that the climate data used in this analysis is a reanalysis product (Daymet) derived from an extensive network of in situ data that are not necessarily representative of entire catchments. In situ data are potentially further biased due to changes in environment or observational methods which negatively impact the stability and homogeneity of these data, which are essential for ensuring accurate and robust trend analysis [39,40]. Due to the scale of this study, it was not possible to ensure the representativeness or homogeneity of the climate data applied, which increases the uncertainty of the analysis.

## 7. Conclusions

To assess the short-term response of landscapes to long-term meteorological trends in mid-latitude North America, we identified trends in the frequency of days of high PET alongside annual water-energy budgets according to a Budyko framework. We found that episodic water and heat stress (HS) has become less frequent in the humid, eastern US, and more common in the arid US as well as in southeastern Canada. These decreases in HS frequency in the eastern US have occurred despite a general increase in daily minimum and maximum temperatures and due to a general increase in annual average PPT. Our findings that streamflow (Q) has been more likely to increase than AET and that changes in AET are uncorrelated with changes in HS frequency indicate that these changes in the water-energy budgets of mid-latitude North America are primarily driven by meteorological forcings as opposed to vegetative response to climate inputs. Finally, we caution readers that the positive finding that HS has become less frequent across much of the eastern US must be contextualized within the Budyko framework, which assumes that landscape vegetation is in equilibrium with a static climate. Therefore, the "decreases" in HS are projections of vegetative response after equilibrium has been achieved, and destructive water and energy stresses are likely to be frequent within these landscapes until such a time as the new equilibrium is achieved. This documentation of long-term trends in the frequency of acute HS may prove useful in identifying (the causes of) long-term trends in vegetative response to climate and help inform management decisions governing land use.

**Funding:** This research received no external funding.

**Institutional Review Board Statement:** Not applicable.

**Informed Consent Statement:** Not applicable.

**Data Availability Statement:** All streamflow data used in this analysis is available from the USGS (https://waterdata.usgs.gov/nwis/rt (accessed on 24 February 2022)) and the Water Office of Canada (https://wateroffice.ec.gc.ca/ (accessed on 24 February 2022)). Watershed boundaries and summary information is available from HydroSHEDS (https://www.hydrosheds.org/ (accessed on 24 February 2022)) and all climate data used in this study is derived from Daymet (https://daymet.ornl.gov/ (accessed on 24 February 2022)). All code used in this article is available from the author at request (arik@climate.ai).

**Conflicts of Interest:** The author declares no conflict of interest.

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
