# Peer review of "Trends in the Frequency of Water and Heat Stress in Mid-Latitude North America since 1980"

_2674-0494, doi:10.3390/meteorology1020009_

Round 1

Reviewer 1 Report

COMMENT TO AUTHORS

Review for Trends in the frequency of water and heat stress in mid-latitude North America since 1980, authored by Tashie et al., 2022.

Some major comments from the reviewer as given below;

The introduction needs serious attention as the starting is not impressive and it seems that the novelty of work is not discussed. It is suggested to discuss the novelty of work by the end of the introduction section. Besides, in 4th paragraph extend in term of in-situ observation and its application over other parts of the globe. For instance, Ullah et al., 2021 uses ground observation against different reanalysis products for assessments of drought events over Pakistan, they found in-situ observation much better as compared to other reanalysis or gridded databases for capturing drought events. https://onlinelibrary.wiley.com/doi/10.1002/joc.7063. Also, move the equation L#70 from the introduction and discuss it in the method section.

L#126: citation is not according to the journal format, same as in L#129, please check throughout the manuscript and bring it according to format.

Figure3: For calculating trends, did the authors remove seasonality from the data? it seems that the author directly used data for calculating trends. If this is the case, removing seasonality is required for trends calculation.

In the discussion section, extend the paragraph in terms of Heatwaves, droughts, hydrological process, and their impacts on other parts of the globe. https://onlinelibrary.wiley.com/doi/10.1029/2021EF002240 https://doi.org/10.1016/j.scitotenv.2021.145327

The conclusion section also needs attention and concluding remarks, to provide some remarks regarding what has been found from the results and their contribution to the community mitigation processes in the future.

Author Response

Thank you for your helpful review. I have incorporated all requested changes as described below. Please note that all of my responses are in italics.

The introduction needs serious attention as the starting is not impressive and it seems that the novelty of work is not discussed. It is suggested to discuss the novelty of work by the end of the introduction section. Besides, in 4th paragraph extend in term of in-situ observation and its application over other parts of the globe. For instance, Ullah et al., 2021 uses ground observation against different reanalysis products for assessments of drought events over Pakistan, they found in-situ observation much better as compared to other reanalysis or gridded databases for capturing drought events. https://onlinelibrary.wiley.com/doi/10.1002/joc.7063. Also, move the equation L#70 from the introduction and discuss it in the method section.

The Introduction has been split into an Introduction and Background section, with the concluding sentence of the Background explicitly addressing the novelty of this research. A description of the uncertainty introduced by the regional scale application of Daymet data (a reanalysis based on an extensive in-situ network) is given on lines 333-343.  The equation has been moved to the methods and recontextualized. And a paragraph describing the limitations of the use of Daymet has been added to lines 335-342

L#126: citation is not according to the journal format, same as in L#129, please check throughout the manuscript and bring it according to format.

Thanks, I tried to identify all citation issues and will be working with the copyeditor to ensure consistency.

Figure3: For calculating trends, did the authors remove seasonality from the data? it seems that the author directly used data for calculating trends. If this is the case, removing seasonality is required for trends calculation.

Apologies for any confusion, Figure 3 represents trends in frequency of HS days per year for each watersheds in the study, plotted against the lat and lon of each watershed. Also please note that all trends are computed against an annual average, thus removing any effect of seasonality.

In the discussion section, extend the paragraph in terms of Heatwaves, droughts, hydrological process, and their impacts on other parts of the globe. https://onlinelibrary.wiley.com/doi/10.1029/2021EF002240 https://doi.org/10.1016/j.scitotenv.2021.145327

Thank you for these reference papers, I have added them and extended the discussion in the Background section

The conclusion section also needs attention and concluding remarks, to provide some remarks regarding what has been found from the results and their contribution to the community mitigation processes in the future.

I have added to the conclusion to describe the potential efficacy of the results.

Thanks so much for your review!

Reviewer 2 Report

Dear Author,

The topic is interesting in the context of the recent global warming and its inpact on environment (e.g. lend vegetation cover including its possibly foodbacks). I have some general comments (additional  ones more specific are in the „pdf“ enclosed).

  • Some discussion on data „qualitity“ and „representativness“ is not quite comlpeted. For example, meteorological „in-situ“ data have not been discussed enough because of rgular networks data have been conducted from them and they are not data from the „first hand“ but useable. It should be careful in discussion on representativnes of in situ data which is dependent on observation environment and time scale. Homogeneiy of data used for trend study is very important (e.g. Pandzic and Likso, 2010; Pandzic at al 2019).

  • Methodology should be described in more details. Good example for description of Budyko Aridity Index was done by Arora (2002). Budyko aridity index is specific because consider „energy“ aspect more expliclity than other aridity/drought/wetness indices. It would be useful to compare, to some extent, that index with others (e.g.  Zargar et al. 2011). In the present description is not clear AET was calculated from Eq. (1) or from Eq. AET=PTT-Q. Also it is not clear how „annual averages“ have been calculated. Treshold criteria for HS is rather arbitrary etc.

  • Seasonality and vegetation feedback should be  described in more details in discussion and more precise in Abstract  and Conclusions.

References:

Arora VK (2002) The use of the aridity indices to acces climate change effect on annual runoff. J. Hydrol., 265, 164-177.

Zargar A, Sadiq R, Naser B, Khan FI (2011) A review of drought indices. Environ Rev 19:333-349 https://doi.org/10.1139/a11-013

Pandžić, K., and Likso,  T.,  2010. Homogeneity of average annual air temperature time series for Croatia. Int. J. Climatol., 30,  1215-1225.

Pandzić, Krešo; Kobold, Mira; Oskoruš, Dijana; Biondic, Božidar; Biondić, Ranko; Bonacci, Ognjen; Likso, Tanja; Curić, Oliver Standard normal homogeneity test as a tool to detect change points in climate-related river discharge variation: case study of the Kupa River Basin // Hydrological sciences journal, 65 (2020), 227-241 doi:10.1080/02626667.2019.1686507

Author Response

Thanks so much for your helpful review of this analysis. I have incorporated all requested changes, as outlined in my response to comments below. Not that my responses are in italics, while the reviewer's comments are not.

The topic is interesting in the context of the recent global warming and its inpact on environment (e.g. lend vegetation cover including its possibly foodbacks). I have some general comments (additional  ones more specific are in the „pdf“ enclosed).

First, from the notes in the pdf:

Comments on lines 8, 12,13, 15, and 18: I have changed the tense to past tense throughout the abstract

Comments on lines 84-85: “substantially” has been removed from the text and discussion of the range of potential thresholds is added to the limitations section. See lines 295-302

Comments on lines 101: thanks for catching this, I mean the R programming language, as described in edits to this line

Comments on line 133: this is a valid point, as I have clarified in the section “Contextualization” on lines 297-302

Comment on line 146: apologies, yes reference AET is defined as AET = PPT – Q, which is then used to calculate the Budyko Curve from EQ1 from which a conceptual AET_potential ~ PET_acutal relationship is derived. Clarified in text

Some discussion on data „qualitity“ and „representativness“ is not quite comlpeted. For example, meteorological „in-situ“ data have not been discussed enough because of rgular networks data have been conducted from them and they are not data from the „first hand“ but useable. It should be careful in discussion on representativnes of in situ data which is dependent on observation environment and time scale. Homogeneiy of data used for trend study is very important (e.g. Pandzic and Likso, 2010; Pandzic at al 2019).

Yes, this is a good point. I have added a discussion of this limitation / source of uncertainty in the section “Contextualization” on lines 314-322. I have also added citations to the two papers listed.

Methodology should be described in more details. Good example for description of Budyko Aridity Index was done by Arora (2002). Budyko aridity index is specific because consider „energy“ aspect more expliclity than other aridity/drought/wetness indices. It would be useful to compare, to some extent, that index with others (e.g.  Zargar et al. 2011). In the present description is not clear AET was calculated from Eq. (1) or from Eq. AET=PTT-Q. Also it is not clear how „annual averages“ have been calculated. Treshold criteria for HS is rather arbitrary etc.

Thank you for bringing this up. I have clarified the calculation of the AI (as noted above). And I have addressed the (in)consistency of the application of different versions of the Budyko framework on lines 131-142. I have also added the Arora reference

Again thanks so much for your helpful review!

Round 2

Reviewer 1 Report

No more comments from side. I applaud the authors and the extent to which they accommodated my suggestions. I believe this manuscript is ready for publication. 

Reviewer 2 Report

Dear Author,

A minor correction is required i.e. on page 11 at line 340 two references "Pandzic and Likso 2010; Pandzic et al. 2020" are cited but not in the reference list.

Zagreb, 28 March 2022